# Temporally-Consistent Survival Analysis

**Lucas Maystre**
Spotify
lucasm@spotify.com

**Daniel Russo**
Columbia University & Spotify
djr2174@gsb.columbia.edu

## Abstract

We study survival analysis in the dynamic setting: We seek to model the time to an event of interest given sequences of states. Taking inspiration from temporal-difference learning, a central idea in reinforcement learning, we develop algorithms that estimate a discrete-time survival model by exploiting a temporal-consistency condition. Intuitively, this condition captures the fact that the survival distribution at consecutive states should be similar, accounting for the delay between states. Our method can be combined with any parametric survival model and naturally accommodates right-censored observations. We demonstrate empirically that it achieves better sample-efficiency and predictive performance compared to approaches that directly regress the observed survival outcome.

## 1 Introduction

Survival analysis provides a statistical framework for reasoning about the time until an event of interest occurs [20, 1]. It is used in a wide range of applications, including modeling patient outcomes in clinical studies [25, 2], predicting customer retention in subscription-based businesses [14, 7], and many more. In the simplest setting, we are given a dataset mapping instances, described by a vector of covariates, to observations about time-to-event. Several aspects distinguish survival analysis from simple regression. The event may not always occur within the observed time-window, leading to so-called *censored* data which contain useful information but need to be treated differently from fully observed data. Furthermore, the focus is often on estimating the rate at which the event occurs, also known as the risk or hazard, as opposed to obtaining a pointwise prediction about the time-to-event.

Across applications, it is becoming increasingly commonplace to collect multiple measurements over time. In this dynamic setting, the data consist of sequences of states instead of a single, static vector of covariates. From an initial state indicating, for instance, features of a patient and a choice of medical treatment, we might observe not just the survival time but rich information on the evolution of their health. For a registered user of a subscription service, we observe not just how long they remain a subscriber but also how their usage evolves over time. We ask the question: Can we take advantage of sequential data to improve survival predictions? Existing work has mostly focused on building representations summarizing measurements up to a given point in time [34, 22], and on modeling the joint distribution of sequences and time-to-event explicitly [35, 17]. In this paper, we contend that there is a different, complementary way to capitalize on dynamic survival data.

We study this problem formally through a discrete-time model. We begin by assuming that sequences of states follow a Markov chain. We model the event of interest as a special terminal state, and we seek to estimate the survival distribution (i.e., the distribution of the hitting time for that terminal state) from any other state. Drawing on ideas from dynamic programming [5], we take advantage of a temporal-consistency condition. The survival distribution at a given state can be written as a combination of *a*) the probability of reaching the terminal state at the next step, and *b*) the survival distribution at the next state. Given full knowledge of the Markov transition matrix, the survival distribution from any state can be computed exactly with a simple recursive algorithm.

36th Conference on Neural Information Processing Systems (NeurIPS 2022).

Building on these insights, we then develop a practical algorithm that fits a parametric survival model by using sample trajectories. Our approach synthesizes temporal-difference learning, a class of algorithms studied in reinforcement learning [33, 6], and survival regression. The algorithm can be used as a drop-in replacement to maximum-likelihood estimation, and, by exploiting temporal consistency, it can dramatically increase statistical precision in certain cases (Section 4.1). We also demonstrate our algorithm's effectiveness on synthetic and real-world datasets (Section 4.2).

Beyond our algorithmic contributions, we hope our presentation may help bridge an intellectual divide between two academic communities. Our primary focus is on making use of the insights underlying temporal difference learning, which is one of the central ideas in reinforcement learning, within survival analysis. But it is also worth noting that many natural problems in reinforcement learning involve survival outcomes, such as the problem of maximizing the length of a session in a recommender system [19]. It is natural to approach such problems in a way that leverages the decades of modeling insight reflected in the survival analysis literature. We sketch such extensions in the supplementary material.

**Organization of the paper.**    In the remainder of this section, we cover related work and set our contributions in context. In Section 2, we introduce key concepts and notation. We then develop our algorithm in Section 3 and discuss connections between RL and dynamic survival analysis. We evaluate our approach empirically on simulated and real-world data in Section 4. Finally, we discuss future work and conclude in Section 5.

## 1.1   Related work

Driven by the growing prevalence of longitudinal datasets, there has been significant interest in the problem of *dynamic prediction* in the literature on survival analysis. Landmarking is a popular method that consists of regressing the survival outcome on all intermediate states, adjusting the regression target based on the time at which the covariates are recorded [34, 27]. The basic idea underpinning landmarking can be used in combination with methods that build (or learn) feature representations that better capture information about a sequence's past up to a given state [34, 29, 22, 13]. Another line of work suggests jointly modeling the sequence dynamics and the survival distribution [35, 17, 3, 22]. Lee and Whitmore [23] view survival analysis by means of a stochastic process reaching a boundary and explicitly consider the implications of the Markov property. Their model is reminiscent of the one we introduce in Section 2. At their core, all these approaches make use of the observed survival outcome as the prediction target. In contrast, we propose changing the prediction target by exploiting a temporal consistency condition. In that sense, our approach is complementary and can be combined with existing dynamic survival models.

The dynamic survival problem we consider in this paper is analogous to the so-called *policy evaluation* problem in reinforcement learning [33]. One of the seminal ideas in the field is temporal-difference learning [32], and our work is considerably influenced by the literature on this topic. The LSTD algorithm of Bradtke and Barto [6] is perhaps the closest work to ours; At a high-level, our algorithm is to survival regression what LSTD is to least-squares regression. We revisit the connections between our work, policy evaluation, and reinforcement learning more generally in Section 3.4. Most policy evaluation schemes estimate a scalar-valued function (the expected return from a state), but some authors introduce methods that estimate a full distribution at each state [24, 4, 12]. In contrast to generic distributional RL algorithms, our work focuses on survival prediction. Our specific focus implies a canonical way of modeling the distribution and a canonical way representing the loss function: The algorithm we introduce in Section 3.3 reveals the central role played by the hazard function.

## 2   Preliminaries

We consider sequences drawn from a Markov chain on a state space $\mathcal{X}$, with initial distribution $\pi_0(x)$ and transition probabilities $p(x'|x)$. We assume that $\mathcal{X}$ contains a special state of interest, which we call the *terminal* state and denote by $\varnothing$. We are interested in estimating the time $T$ until the sequence reaches the terminal state from any initial state $x_0 \neq \varnothing$. In the literature on stochastic processes, this is commonly referred to as the hitting time of the terminal state [26, 23]. In this paper, we use the terminology of survival analysis and refer to $T$ as the *time-to-event*. We call the

probability distribution of $T$ the *survival distribution*. We describe a survival distribution by using three interrelated functions:

$$
\begin{aligned}
\text{probability mass function} && f(k|x) &= \mathbf{P}[T = k \mid x_0 = x], \\
\text{survival function} && S(k|x) &= \mathbf{P}[T > k \mid x_0 = x], \\
\text{hazard function} && h(k|x) &= \mathbf{P}[T = k \mid T \geq k, x_0 = x].
\end{aligned}
$$

It is possible to express any one function in terms of any other, for example $S(k|x) = \prod_{k'=1}^{k}[1 - h(k'|x)]$. With a slight abuse of notation, we let $f(k|\varnothing) = S(k|\varnothing) = 0$ for $k \geq 1$, and

$$
h(0|x) = f(0|x) = \begin{cases} 1 & \text{if } x = \varnothing, \\ 0 & \text{otherwise,} \end{cases} \qquad S(0|x) = \begin{cases} 0 & \text{if } x = \varnothing, \\ 1 & \text{otherwise.} \end{cases}
$$

In the remainder of the paper, we estimate survival up to some finite horizon $K$. That is, we are interested in the hazard probabilities $h(1|x), \ldots, h(K|x)$, or equivalently, in $f(1|x), \ldots, f(K|x)$, with $S(K|x)$ capturing the remaining probability mass.

**Datasets of sequences.** In practice, we will be estimating survival from sequences sampled from the Markov chain. We assume that each sequence $s = (x_0, x_1, \ldots, x_t)$ that we observe either *a)* ends as soon as it reaches the terminal state, or *b)* has not yet reached the terminal state. A sequence where $x_t \neq \varnothing$ is called *right-censored*. We call $c = \mathbf{1}\{x_t \neq \varnothing\}$ the censoring indicator, and the index $t$ of the last observed state is the time-to-event or censoring time. We collect observed sequence into a dataset $\mathcal{D} = \{s_n : n = 1, \ldots, N\}$, noting that sequences can be of different lengths. Given such a dataset, a natural choice for the horizon $K$ is the length the longest sequence.

## 2.1 Survival regression

In order to achieve better predictive performance given finite data, we often consider parametric approximations to the survival distribution. Generally, the hazard is the most natural quantity to model. Assuming that each state $x$ is described by a feature vector $\phi_x \in \mathbf{R}^D$, we approximate the hazard by a function $h_\theta(k|x)$, parametrized by $\theta$, that is a function of $x$ through $\phi_x$. Intuitively, if two states $x, x'$ share similar state representations (i.e., $\phi_x \approx \phi_{x'}$), then the two states should have similar survival distributions. Specific parametrization choices might link together the survival probabilities across states, across time, or both.

The discrete-time Cox proportional-hazards (PH) model [8] is widely used in practice. Letting $\theta = (\alpha, \beta)$ where $\alpha \in \mathbf{R}^K$ and $\beta \in \mathbf{R}^D$, the Cox PH model is given by

$$
\text{logit}[h_\theta(k|x)] = \beta^\top \phi_x + \alpha_k,
$$

where $\text{logit}(p) = \log[p/(1-p)]$. The elements of $\alpha$ correspond to the baseline (state-independent) hazard log-odds, and the vector $\beta$ describes the (time-independent) effect of the features. This model can be extended, e.g., by replacing the linear dependence on $\phi_x$ with a deep neural network [21]. As an alternative to the Cox PH model, some authors suggest making use of a flexible distribution on the positive integers such as the beta-geometric, and rewrite the parameters of the distribution as linear or non-linear functions of $\phi_x$ [16, 18].

Given a dataset $\mathcal{D} = \{s_n\}$, we can learn a parametric approximation by maximizing the likelihood of the model under the data. The maximum-likelihood estimator (MLE) can be cast as a weighted binary classification problem with cross-entropy loss [20, 9],

$$
\theta_{\text{MLE}} \in \arg\min_\theta \sum_{n=1}^{N} \sum_{k=1}^{K} w_{nk} H[y_{nk} \| h_\theta(k|x_{n0})], \tag{1}
$$

where $H[p\|q] = -p \log q - (1-p) \log(1-q)$ is the binary cross-entropy, $w_{nk} = \mathbf{1}\{t_n \geq k\}$, and the target $y_{nk} = \mathbf{1}\{t_n = k \wedge c_n = 0\}$ is an indicator variable for the event. The MLE only depends on triplets of the form $(x_0, t, c)$, i.e., it completely ignores intermediate states. Landmarking extends (1) by effectively augmenting the data to include triplets of the form $(x_\ell, t - \ell, c)$ for $\ell < t$.[1]

---

[1]Making a parallel to reinforcement learning methods, this is similar to *any-visit Monte Carlo* methods [33].

# 3 Methodology

In this section, we develop efficient algorithms that estimate the survival distribution from any state. Our approach takes advantage of the following key idea. By using the law of total probability and the Markov property, we can write functions characterizing the survival distribution as recursions. For all $k \geq 1, x \in \mathcal{X}$, we have

$$
\begin{aligned}
f(k|x) &= \mathbf{E}_{x' \sim p(\cdot|x)}[f(k-1|x')], \\
S(k|x) &= \mathbf{E}_{x' \sim p(\cdot|x)}[S(k-1|x')].
\end{aligned}
\tag{2}
$$

Intuitively, these identities capture a notion of temporal consistency. For example, the second one states that the probability of surviving $k$ steps starting from a given state should be equal to the probability of surviving $k-1$ steps starting from the next state, on average.

We proceed in three steps. We begin by assuming that the Markov chain dynamics are known and present a dynamic programming approach to computing the exact survival distribution from every state (Section 3.1). Then, we extend the approach to fit a parametric approximation to the survival distribution (Section 3.2). Finally, we consider the setting where we only have access to a finite dataset of sequences, and we design a practical algorithm (Section 3.3). We discuss connections to RL and extensions in Section 3.4.

For simplicity, we assume that the state space is finite and has cardinality $|\mathcal{X}| = X$. We do so for didactic purposes, and in practice this assumption is not required to use the algorithm we present in Section 3.3.

## 3.1 Exact solution via dynamic programming

We denote the horizon-$K$ survival distribution from state $x$ as the $(K+1)$-dimensional probability vector

$$
q_x^\star = [f(1|x) \quad \cdots \quad f(K|x) \quad S(K|x)],
$$

where $S(K|x) = 1 - \sum_{k=1}^{K} f(k|x)$. Given any model $\tilde{f}(k|x)$ (not necessarily matching the true probability mass function) and the corresponding vector $q_x = [\tilde{f}(1|x) \quad \cdots \quad \tilde{f}(K|x) \quad \tilde{S}(K|x)]$, we define the backshift operator $\overleftarrow{\cdot}$ as

$$
\overleftarrow{q_x} = [\tilde{f}(0|x) \quad \cdots \quad \tilde{f}(K-1|x) \quad \tilde{S}(K-1|x)],
$$

where $\tilde{f}(0|x) = \mathbf{1}\{x = \varnothing\}$ and $\tilde{S}(K-1|x) = \tilde{f}(K|x) + \tilde{S}(K|x)$. Finally, we define the survival matrix $Q^\star$ as the $X \times (K+1)$ row-stochastic matrix obtained by stacking rows $q_x^\star$, and the transition matrix $P$ as the as the $X \times X$ matrix such that $p_{xx'} = p(x'|x)$. Equipped with these definitions, we can write a variant of (2) in matrix form as

$$
Q^\star = P\overleftarrow{Q^\star}.
$$

This matrix identity suggests the following iterative scheme. Starting with a matrix $Q_0$ formed by an initial guess for the time-to-event probabilities $f_0(k|x)$, we can iteratively compute

$$
Q_{i+1} = P\overleftarrow{Q_i},
\tag{3}
$$

The following proposition shows that applying this mapping $K$ times recovers the true survival distribution $Q^\star$, no matter what the starting point $Q_0$ is.

**Proposition 1.** *For any initial survival distribution $f_0(k|x)$ and corresponding row-stochastic matrix $Q_0$, the sequence $(Q_i)_{i=0}^\infty$ defined by (3) converges to $Q^\star$ after $K$ steps.*

*Sketch of proof.* We show, by induction on $i$, that the first $i$ columns of $Q_i$ match those of $Q^\star$. As all survival matrices are row-stochastic, their $K + 1$ columns must add up to exactly one, and $K$ iterations are sufficient to ensure that $Q_K = Q^\star$. $\qquad\square$

## 3.2 Approximate solution with a parametric model

So far, we have represented the survival distributions exactly, at every state, by letting the survival matrix $Q$ range freely over all row-stochastic matrices. Alternatively, we might want to approximate the survival distribution at $x$ by constraining it to a parametric form

$$q_{\theta,x} = [f_\theta(1|x) \quad \cdots \quad f_\theta(K|x) \quad S_\theta(K|x)]$$

such as the Cox PH model described in Section 2.

In this case, we propose to modify (3) by projecting each iterate onto the set $\mathcal{Q} = \{Q_\theta : \theta \in \Theta\}$ of all feasible survival matrices. Given a distribution over states $\pi(x)$, we define the projection operator $\Pi$ to be a function from $\mathbf{R}^{X \times (K+1)}$ to $\mathcal{Q}$ satisfying

$$\Pi Q \in \underset{Q_\theta \in \mathcal{Q}}{\arg\min}\, d_\pi(Q, Q_\theta), \qquad\qquad d_\pi(Q, Q') = \sum_x \pi(x) H[q_x \| q'_x], \qquad (4)$$

where we overload $H$ to denote the categorical cross-entropy $H[u\|v] = -\sum_{k=1}^{K+1} u_k \log v_k$. Informally, $\Pi Q$ is a feasible matrix that is closest to $Q$ in a weighted cross-entropy sense. In the information geometry literature, $\Pi Q$ is known as the *reverse information projection* of $Q$ onto $\mathcal{Q}$ and is closely related to maximum-likelihood estimation [10, 11]. Starting from any $Q_0 \in \mathcal{Q}$, we refine a parametric estimate to the survival matrix iteratively as

$$Q_{i+1} = \Pi P \overleftarrow{Q}_i. \qquad (5)$$

Whether (5) converges or not remains an open question in the general case, but we identify a class of parametric families where convergence is guaranteed.

**Assumption A1** (Separable log-linear model). *The parameter set factors as $\Theta = \Theta^{(1)} \times \cdots \times \Theta^{(K)}$ where each $\Theta^{(k)}$ is convex. For each $x, k$ there exists a vector $\phi_{x,k}$ such that $\mathrm{logit}(h_\theta(k|x)) = \phi_{x,k}^\top \theta^{(k)}$ depends on $\theta = (\theta^{(1)}, \cdots, \theta^{(K)})$ only through $\theta^{(k)}$. Assume $\sum_x \pi(x)\phi_{x,k}\phi_{x,k}^\top$ has full rank for all $k$.*

**Assumption A2** (Non-degeneracy). *The population MLE $\arg\min_\theta d_\pi(Q^\star, Q_\theta)$ exists.*

Intuitively, a separable model constrains the survival distribution across states but not across time. To understand our second assumption, note that instances of our problem with $K = 1$ are similar to logistic regression with soft-labels. Logistic regression has degenerate behavior in problems where the data are perfectly separable. We rule that out implicitly by assuming a finite MLE.

**Proposition 2.** *If A1–A2 are satisfied, then, for any initial row-stochastic matrix $Q_0$, the sequence $(Q_i)_{i=0}^\infty$ defined by (5) converges to a unique fixed point $\bar{Q}$ after $K$ steps.*

*Sketch of proof.* We start by rewriting the projection operator in terms of the hazard function $h_\theta$. We show by induction on $i$ that, for models satisfying A1–A2, there is a unique $\bar{\theta}$ such that $h_{\theta_i}(k|x) = h_{\bar{\theta}}(k|x)$ for all $k \leq i$ and all $x$. Given that $Q_i$ is a function of $h_{\theta_i}$, we conclude that there is a fixed point $\bar{Q}$ such that $Q_i = \bar{Q}$ for all $i \geq K$. $\qquad\square$

Note that $\bar{Q}$ minimizes the temporal inconsistency $d_\pi(\bar{Q}, P\overleftarrow{Q})$, which might be different from minimizing the divergence from the true survival matrix $Q^\star$.

## 3.3 Temporally-consistent survival regression

In most applications, we usually do not have full access to the transition probabilities $p(x'|x)$. Instead, we only get to observe a dataset of sequences $\mathcal{D} = \{s_n : n = 1, \ldots, N\}$ generated as per Section 2. Building on the ideas developed above, we introduce *temporally-consistent survival regression* (TCSR), a practical algorithm that estimates a survival model from samples. We describe the procedure in Algorithm 1.

The algorithm takes as input pairs of successive states $\mathcal{T} = \{(x_m, x'_m)\}$ that we obtain by breaking down the $N$ sequences into $M = \sum_n t_n$ one-step transitions. Starting with an initial guess for $\theta$, the algorithm then repeatedly solves an optimization problem until convergence. It is instructive to compare the optimization problem on Line 8 to the maximum-likelihood estimator (1). Similarly to

---

**Algorithm 1** Temporally-consistent survival regression (TCSR).

---

**Require:** dataset of transitions $\mathcal{T} = \{(x_m, x'_m)\}$, initial parameters $\theta$.
 1: **repeat**
 2:     **for** $m = 1, \ldots, M$ **do**
 3:         $y_{m1} \leftarrow \mathbf{1}\{x'_m = \varnothing\}$                      $\triangleright$ Based on one-step observed outcome.
 4:         $w_{m1} \leftarrow 1$
 5:         **for** $k = 2, \ldots, K$ **do**                       $\triangleright$ Based on predictions at $x'_m$.
 6:             $y_{mk} \leftarrow h_\theta(k-1|x'_m)$
 7:             $w_{mk} \leftarrow S_\theta(k-2|x'_m)$
 8:     $\theta \leftarrow \arg\min_\theta \sum_{m=1}^{M} \sum_{k=1}^{K} w_{mk} H[y_{mk} \| h_\theta(k|x_m)]$
 9: **until** $\theta$ has converged

---

the MLE, our approach casts survival estimation as a weighted binary classification problem. Whereas the MLE regresses hard binary targets that are exclusively based on the observed time-to-event or censoring time, we regress a combination of hard targets (whether the sequence terminates at the next step, Line 3) and soft targets (predictions at the next state, Line 6). Similarly, our weights are functions of observations and predictions at the next state.

The connection to the MLE has important implications in practice. It means that we can implement TCSR by reusing existing code for survival regression or binary classification. At each iteration, we first compute weights and pseudo-targets based on a current estimate of the model, and we can then delegate the optimization problem on Line 8 to existing off-the-shelf software [9]. It also suggests a simple heuristic to reason about the running time: Given existing code that computes the MLE in time $\tau$, we expect $L$ iterations of TCSR to take time $\approx (M/N)L\tau$.

In addition to the perspective provided by the MLE, it is also possible to view TCSR through the lens of the approximate dynamic programming scheme we have developed in Section 3.2. In fact, TCSR is identical to (5) on a certain empirical Markov chain, as made precise by the following proposition.

**Proposition 3.** *Algorithm 1 is equivalent to the fixed-point iteration* (5) *with the empirical Markov chain and state distribution*

$$p(x'|x) = \sum_m \mathbf{1}\{x_m = x, x'_m = x'\} / \sum_m \mathbf{1}\{x_m = x\}, \qquad \pi(x) = \sum_m \mathbf{1}\{x_m = x\}/M.$$

*Sketch of proof.* A key ingredient consists of recasting the projection operator $\Pi$ in (4) into a weighted binary classification problem similar to that of Line 8 in Algorithm 1. By using the fact that $f(k|x) = h(k|x)S(k-1|x)$ and $S(k|x) = \prod_{k'=1}^{k}[1 - h(k'|x)]$, we can rewrite the categorical cross-entropy from a distribution $z$ to $q_{x,\theta}$ in terms of binary cross-entropy to the hazard probabilities:

$$
\begin{aligned}
H[z\|q_{x,\theta}] &= -\sum_{k=1}^{K} z_k \log f_\theta(k|x) - z_{K+1} \log S_\theta(K|x) \\
&= -\sum_{k=1}^{K} z_k \log h_\theta(k|x) - \sum_{k=1}^{K+1} z_k \sum_{k'=1}^{k-1} \log[1 - h_\theta(k'|x)] \\
&= -\sum_{k=1}^{K} \left\{ z_k \log h_\theta(k|x) - \left(\sum_{k'=k+1}^{K+1} z_{k'}\right) \log[1 - h_\theta(k|x)] \right\} \\
&= \sum_{k=1}^{K} w_k H[y_k \| h_\theta(k|x)],
\end{aligned}
$$

where $w_k = \sum_{k'=k}^{K+1} z_{k'}$ and $y_k = z_k/w_k$. If $z = \overleftarrow{q'_{x'}}$, we find that $y_k = h'(k-1|x')$ for $k \geq 1$, $w_1 = 1$, and $w_k = S'(k-2|x')$ for $k \geq 2$. The remainder of the proof relies on a linearity property of the cross-entropy. $\qquad\square$

This connection is helpful to analyze TCSR. For example, the following result is an immediate corollary of Propositions 2 and 3.

**Corollary 4.** *If $\mathcal{Q}$ satisfies A1–A2 on the empirical Markov chain defined in Proposition 3, then, for any initial parameters $\theta_0$, the sequence $(\theta_i)_{i=0}^{\infty}$ defined by successive iterations of Algorithm 1 converges to a unique fixed point $\bar{\theta}$ after $K$ iterations.*

Note that many parametric families used to model survival in practice do not satisfy the assumptions we require to guarantee convergence. In particular, they are not separable across time and therefore do not satisfy A1. Nevertheless, our experiments suggest that TCSR is very stable in practice, and we have not encountered a single case where it failed to converge (see also Section 4.2).

**Continuous state spaces.** We began Section 3 by assuming that the state space $\mathcal{X}$ is finite. With Algorithm 1, however, this assumption is no longer necessary. In practice, by using smooth parametric functions, TCSR works with continuous state representations without any modifications.

### 3.4 Connections to RL & extensions

Our presentation synthesizes temporal difference learning—one of the central intellectual ideas in RL—and survival analysis. The consistency equations (2) that form the basis of our approach are closely related to the Bellman equations in Markov decision processes [33, Sec. 3.5]. The fixed-point iterations (3) and (5) mirror (approximate) dynamic programming approaches to policy evaluation [5, Sec. 2.2].

There are also striking parallels between TCSR and the LSTD algorithm of Bradtke and Barto [6]. Both algorithms exploit temporal consistency by linking predictions at successive states, and they both solve a sequence of maximum-likelihood problems on pseudo-observations. They can both be recast as fixed-point iterations on an empirical Markov chain. Whereas LSTD learns a scalar-valued function that minimizes a squared loss, TCSR learns a distribution-valued function that minimizes a cross-entropy loss. LSTD can be generalized to account for observations more than one state ahead, a variant called LSTD($\lambda$). We can generalize TCSR in a similar way, providing a unified perspective on Algorithm 1 and the MLE (1). We sketch such a generalization in Appendix A.3.

In RL, the algorithms SARSA and Q-learning naturally extend the insight of temporal-difference learning to problems with actions [33]. We can provide a similar generalization and learn survival distributions that depend on state-action pairs. By interleaving this estimation problem with a policy improvement step, we end up with a policy iteration algorithm that is tailored to survival problems. We explore these extensions in Appendix A.3.

## 4 Experimental evaluation

In this section we evaluate TCSR (Algorithm 1) empirically on synthetic and real-world data. Our main focus is data-efficiency: We seek to validate that we can exploit temporal consistency to learn better models with fewer samples. Throughout the section we approximate the survival distribution by learning the parameters $\theta = (\alpha, \beta)$ of a Cox PH model

$$\text{logit}[h_\theta(k|x)] = \beta^\top \phi_x + \alpha_k,$$

where $\phi_x$ is a feature vector describing state $x$. Restricting our attention to this model lets us focus on the impact of the learning algorithm, independently of the impact of the parametric form of the model itself. We compare TCSR to two alternative approaches to learning a survival model.

**Initial state.** We regress the observed outcome directly on the initial state of the sequence, by maximizing the log-likelihood (1). This approach corresponds the static setting, as it only employs triplets of the form $(x_0, t, c)$ and ignores information contained in the sequence of states. If the sequences of features $(\phi_{x_0}, \phi_{x_1}, \ldots)$ are not Markov, this approach might produce models that are less biased.

**Landmarking.** We augment the dataset used in the initial state approach by including every intermediate state and the remaining time-to-event or time-to-censoring, i.e., triplets of the form $(x_\ell, t - \ell, c)$ for $\ell < t$. Parameter inference is carried out by maximizing the likelihood under the augmented dataset. This is the dominant approach to dynamic survival prediction in the biostatistics literature [34, 27].

We provide a software library with a reference implementation of TCSR in the Python programming language as well as computational notebooks that enable reproducing the results presented in this section at `https://github.com/spotify-research/tdsurv`.

### 4.1 Statistical benefits of temporal consistency

We begin by providing a concrete generative model that illustrates the benefits of exploiting temporal consistency (Figure 1, left). In this problem, we seek to determine the impact on survival of $L$ different treatments, corresponding to $L$ different initial states. For each treatment, we observe exactly

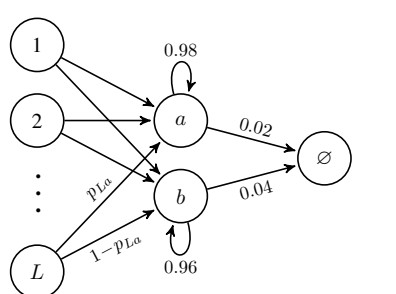
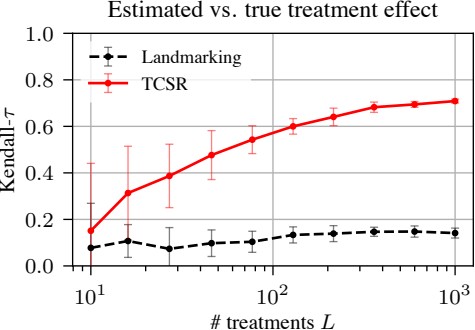

Estimated vs. true treatment effect

Figure 1: Example illustrating the statistical benefits of exploiting temporal consistency. *Left*: Markov chain generating the data. *Right*: performance of two estimators as the number of treatments grows.

10 sequences starting at the corresponding state; sequences that do not reach the terminal state within 5 steps are right-censored. Starting from a given initial state $x$, the sequence transitions to state $a$ (healthy) with probability $p_{xa}$, and to state $b$ (sick) with probability $1 - p_{xa}$. From states $a$ and $b$, time-to-event is geometrically distributed with probability 0.02 and 0.04, respectively. As such, the survival distribution at each of the treatment states is entirely determined by the transition probabilities $\{p_{xa}\}$, which induce a natural ranking over treatments. In a simple two-stage problem like this, transitions to state $a$ or $b$ are called *surrogate outcomes* [28]. We set $\phi_x$ to be the one-hot encoding of the state identifier and learn Cox-PH models by using TCSR and landmarking.

In Figure 1 (right) we report the correlation between the rankings induced by $\{p_{xa}\}$ and by the model coefficients $\{\beta_x\}$ for increasing values of $L$, averaged over 10 realizations. We observe that TCSR is able to discriminate between the treatments increasingly better as $L$ grows, despite the number of sequences per treatment remaining constant. Intuitively, as $L$ grows, we are able to learn increasingly accurate models of survival from states $a$ and $b$ by pooling data from all sequences. By leveraging the Markov assumption and regressing *predictions* at $a$ and $b$ on the initial states, we remove a large part of the statistical noise in the individual sequences, thus improving our ability to identify the effect of each treatment on survival. Approaches that directly regress the observed time-to-event outcome, such as landmarking, are fundamentally unable to make use of this data-pooling advantage.

## 4.2 Empirical performance on three datasets

Next, we consider three dynamic survival datasets and evaluate the performance of models on two tasks of practical interest.

1. Rank instances by predicted time-to-event based on the initial state of the sequence, $x_0$. To this end, we score sequences by the hazard log-odds $\beta^\top \phi_{x_0}$ and measure the concordance index (CI, see Appendix B.1 for a precise definition). Random scores achieve a CI of 0.5, while a CI of 1 corresponds to perfect agreement between scores and observed time-to-event.

2. Estimate the effect of covariates on the time-to-event. In the Cox PH model, recall that the regression coefficient $\beta_i$ captures the linear effect of the $i$th covariate on the hazard log-odds. We evaluate a model by measuring the root-mean-squared error (RMSE) between the regression coefficients $\beta$ and gold-standard coefficients $\beta^\star$.

These two tasks are particularly relevant to clinical applications. For example, accurately ranking instances by time-to-event could help identify patients which are most at risk early on. In Appendix B.3, we provide results on two additional metrics that capture a model's ability to make accurate probabilistic predictions, the predictive log-likelihood and the integrated Brier score [15].

Of the three datasets we consider, two contain real data from clinical studies and are publicly available online [30]. The first study tracks survival outcomes from 312 patients diagnosed with PBC, a rare liver disease [25]. Patient features include static information such as gender and treatment, and 12 dynamic biomarkers that are measured at study entry and at follow-up visits. The second study follows 467 patients diagnosed with HIV / AIDS [2]. In this study, a single dynamic biomarker, the CD4 cell count, is measured repeatedly. The two studies span multiple years, and follow-up visits occur at a regular cadence. We discard the time between successive visits and simply consider the

Table 1: Summary statistics of three datasets studied in Section 4.2.

| Dataset | $|\mathcal{D}|$ | Median length | Max length | # static feat. | # dyn. feat. | Frac. censored |
|---------|-----------------|---------------|------------|----------------|--------------|----------------|
| PBC2 [25] | 312 | 5 | 16 | 3 | 12 | 55.1% |
| AIDS [2] | 467 | 3 | 5 | 4 | 1 | 59.7% |
| RW | — | 4 | 11 | 0 | 20 | 21.9% |

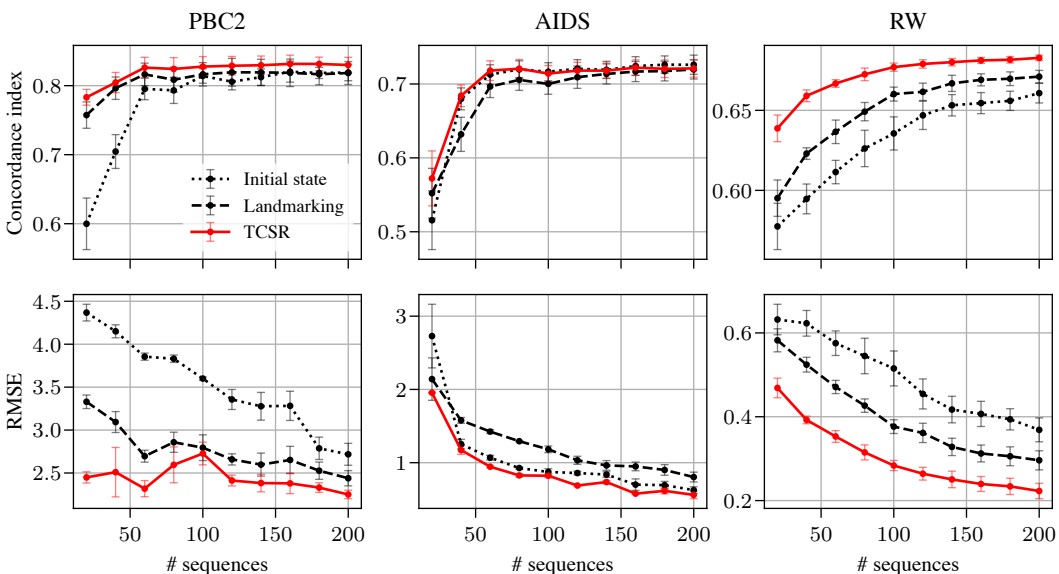

Figure 2: Empirical performance of learning algorithms on three datasets. We report the mean and standard deviation over 5 splits. Concordance index: higher is better, RMSE: lower is better.

discrete-time sequences induced by successive measurements. We complement these clinical datasets with a synthetic dataset based on a 20-dimensional random walk model. At each step, the sequence terminates with a probability that depends on the location in 20D space. Summary statistics for all datasets are presented in Table 1, and more details are provided in Appendix B.2.

The feature vectors $\{\phi_x\}$ are obtained by concatenating the static controls with the dynamic features. We define the gold-standard coefficients $\beta^\star$ as the MLE (1) under the full dataset. We train models on increasingly larger subsets of the data. We report the average performance obtained by using 5-fold cross-validation in Figure 2.

We observe that landmarking and TCSR, the two methods that make use of the full sequences, outperform the static baseline on PBC2 and RW. On AIDS, landmarking performs poorly, while TCSR performs similarly (in terms of CI) or slightly better (in terms of RMSE) than initial state. Recall that this dataset has a single dynamic feature, and thus the information contained in the sequence dynamics is likely limited. Overall, TCSR generalizes systematically better than landmarking, e.g., necessitating less than $4\times$ fewer samples to reach a given performance level on RW and converging to a strictly better model on PBC2. These findings suggest that the statistical benefits illustrated in Section 4.1 extend to real-world data and lead to meaningful improvements to predictive performance.

**Running time and convergence.** In our implementation of TCSR, we delegate Line 8 in Algorithm 1 to a Newton-CG solver provided by the SciPy library [31]. For simplicity, we have fixed the number of iterations to 30 across all datasets and experiments, but the algorithm often converges in fewer iterations. Interestingly, the actual running time of TCSR is usually significantly less than $30\times$ the running time of landmarking. We believe that this is because initializing the inner optimization (Line 8) with the previous iterate accelerates the convergence of the Newton-CG optimizer substantially.

# 5 Conclusion & future work

Driven by insights from temporal-difference learning, we have introduced a new algorithm that estimates a survival model in the dynamic-data setting. Our algorithm takes advantage of a temporal-consistency condition that we know, a priori, should be satisfied by the true survival distribution. Our approach be can combined with existing dynamic survival models and inference software, and it only requires changing the regression targets. As such, we believe that our approach will be of interest to researchers and practitioners alike.

We envision two broad directions for future work. The first direction consists of extending our algorithmic approach to accommodate a wider range of use-cases. We believe that the extension to the *competing risks* setting [1] is straightforward, and would simply entail extending the binary problem (surviving or not at each step) to a multiclass problem reflecting the multiple possible outcomes. The extension to irregularly-sampled, *continuous-time* sequences is also worth considering. Given a continuous-time, stationary Markov process $x(t)$, we can write a temporal-consistency condition for any time interval $\Delta > 0$ as

$$S(t|x) = \mathbf{E}_{x' \sim p_\Delta(\cdot|x)}[S(t - \Delta|x')],$$

where $p_\Delta(\cdot|x)$ is the probability density function of $x(t + \Delta)$ conditional on $x(t) = x$. We have preliminary ideas on how to turn this into a practical algorithm, but more research is needed. A second direction is to make use of our approach to address sequential decision-making problems that are naturally expressed in terms of maximizing survival. This requires extending our developments to encompass problems with actions, a generalization we sketch in Appendix A.3.

**Acknowledgments.** We thank Ksenia Konyushkova for feedback on early versions of the idea presented in this paper. We also thank Mounia Lalmas and the anonymous reviewers for their careful review and proofreading.

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
