# Temporally-Consistent Survival Analysis
# Supplementary Material

**Lucas Maystre**
Spotify
lucasm@spotify.com

**Daniel Russo**
Spotify & Columbia Business School
djr2174@gsb.columbia.edu

## A  Methodology

This appendix is organized as follows. In Section A.1, we provide complete proofs for the results presented in Section 3 of the main text. We develop a generalization of TCSR that considers multi-hop transitions in Section A.2. Finally, in Section A.3, we revisit connections with RL and sketch extensions of our approach to problems with actions.

### A.1  Proofs

For convenience, we briefly recall each result before presenting a complete proof. We start with the fixed-point iteration that recovers the exact finite-horizon survival distribution $Q^\star$.

**Proposition 1.** *For any initial survival distribution $f_0(k|x)$ and corresponding row-stochastic matrix $Q_0$, the sequence $(Q_i)_{i=0}^\infty$ defined by (3) converges to $Q^\star$ after $K$ steps.*

*Proof.* We will show, by induction on $i$, that the first $i$ columns of $Q_i$ match those of $Q^\star$. As all survival matrices are row-stochastic, their $K+1$ columns must add up to exactly one, and $K$ iterations are sufficient to ensure that $Q_K = Q^\star$.

Denote the row of matrix $Q_i$ corresponding to state $x$ by the vector $[f_i(1|x) \quad \cdots \quad f_i(K|x) \quad S_i(K|x)]$. By definition of the backshift operator and of the function $f_j(0|\cdot)$, we have

$$Q_1 = P\overleftarrow{Q_0} \implies f_1(1|x) = \sum_{x'} p(x'|x)f_0(0|x') = \sum_{x'} p(x'|x)\mathbf{1}\{x' = \varnothing\} = p(\varnothing|x)$$
$$\implies \mathrm{col}_1(Q_1) = \mathrm{col}_1(Q^\star).$$

Next, fix $i$ and assume that the first $\min\{i, K\}$ columns of $Q_i$ match those of $Q^\star$. This implies $f_i(k|x) = f(k|x)$ and $S_i(k|x) = S(k|x)$ for all $x$ and all $k \le \min\{i, K\}$. Making use of the identities (2), we obtain

$$Q_{i+1} = P\overleftarrow{Q_i} \implies \begin{cases} f_{i+1}(k|x) = \sum_{x'} p(x'|x)f_i(k-1|x') = \sum_{x'} p(x'|x)f(k-1|x') = f(k|x'), \\ S_{i+1}(k|x) = \sum_{x'} p(x'|x)S_i(k-1|x') = \sum_{x'} p(x'|x)S(k-1|x') = S(k|x') \end{cases}$$
$$\implies \mathrm{col}_k(Q_{i+1}) = \mathrm{col}_k(Q^\star),$$

for all $k \le 1 + \min\{i, K\}$, completing the proof. $\square$

Next, we look at the convergence of the fixed-point iteration that projects iterates onto a parametric family.

**Proposition 2.** *If A1–A2 are satisfied, then, for any initial row-stochastic matrix $Q_0$, the sequence $(Q_i)_{i=0}^\infty$ defined by (5) converges to a unique fixed point $\bar{Q}$ after $K$ steps.*

36th Conference on Neural Information Processing Systems (NeurIPS 2022).

*Proof.* By using the fact that $f(k|x) = h(k|x)S(k-1|x)$ and $S(k|x) = \prod_{k'=1}^{k}[1 - h(k'|x)]$, we can rewrite the divergence $d_\pi(Q, Q_\theta)$ in terms of binary cross-entropy to the hazard probabilities. Letting $Q = [q_{x,k}]$, we have

$$
\begin{aligned}
d_\pi(Q, Q_\theta) &= \sum_x \pi(x) \left( -\sum_{k=1}^{K} q_{x,k} \log f_\theta(k|x) - q_{x,K+1} \log S_\theta(K|x) \right) \\
&= \sum_x \pi(x) \left( -\sum_{k=1}^{K} q_{x,k} \log h_\theta(k|x) - \sum_{k=2}^{K+1} q_{x,k} \sum_{k'=1}^{k-1} \log[1 - h_\theta(k'|x)] \right) \\
&= \sum_x \pi(x) \left( -\sum_{k=1}^{K} \left\{ q_{x,k} \log h_\theta(k|x) - \left( \sum_{k'=k+1}^{K+1} q_{x,k'} \right) \log[1 - h_\theta(k|x)] \right\} \right) \\
&= \sum_x \pi(x) \left( \sum_{k=1}^{K} w_{xk} H[y_{xk} \| h_\theta(k|x)] \right) \\
&= \sum_{k=1}^{K} \left( \sum_x \pi(x) w_{xk} H[y_{xk} \| h_\theta(k|x)] \right),
\end{aligned}
$$

where $w_{xk} = \sum_{k'=k}^{K+1} q_{x,k'}$ and $y_{xk} = q_{x,k}/w_{xk}$. If $Q = P\overleftarrow{Q_{\theta'}}$, it is easy to verify that

$$
y_{xk}(\theta') = \sum_{x'} p(x'|x) h_{\theta'}(k-1|x'),
$$

$$
w_{xk}(\theta') = \begin{cases} \sum_{x'} p(x'|x) \prod_{\ell=0}^{k-2}[1 - h_{\theta'}(\ell|x')] & \text{if } k \geq 2, \\ 1 & \text{if } k = 1, \end{cases}
$$

where we make the dependence of $y_{xk}$ and $w_{xk}$ on parameters $\theta'$ explicit. This reformulation of the divergence reveals two important properties of the projection operator for separable models.

1. Minimizing $d_\pi(Q, Q_\theta)$ with respect to $\theta$ can be decomposed into minimizing $K$ independent quantities that are functions of the hazard at each step.

2. The pseudo-weights and pseudo-labels $w_{xk}(\theta)$ and $y_{xk}(\theta)$ depend on $\theta$ only through $(\theta^{(1)}, \ldots, \theta^{(k-1)})$.

Let $\theta_i = (\theta_i^{(1)}, \ldots, \theta_i^{(K)})$ be the parameters corresponding to $Q_i$. We will now show by induction on $i$ that, for all $i$ and all $k \leq \min\{i, K\}$, $\theta_i^{(k)}$ exists, is unique, and is equal to $\theta_k^{(k)}$. For $i = 1$, we have

$$
\theta_1 \in \arg\min_\theta d_\pi(P\overleftarrow{Q_0}, Q_\theta) \implies \theta_1^{(1)} \in \arg\min_\theta \sum_x \pi(x) w_{x1} H[y_{x1} \| h_\theta(1|x)],
$$

where $w_{x1} = 1$ and $y_{x1} = \sum_{x'} p(x'|x)\mathbf{1}\{x' = \varnothing\} = p(\varnothing|x)$. Thus, $\theta_1^{(1)}$ is identical to the population MLE, which, as a consequence of A1–A2, exists and is unique. Next, fix $i$ and assume that $\theta_i^{(k)} = \theta_k^{(k)}$ exists and is unique for all $k \leq \min\{i, K\}$. Then

$$
\begin{aligned}
\theta_{i+1} &\in \arg\min_\theta d_\pi(P\overleftarrow{Q_i}, Q_\theta) \\
&\implies \theta_{i+1}^{(k)} \in \arg\min_\theta \sum_x \pi(x) w_{xk}(\theta_i) H[y_{xk}(\theta_i) \| h_\theta(k|x)] \quad \forall k \leq i \\
&\implies \theta_{i+1}^{(k)} \in \arg\min_\theta \sum_x \pi(x) w_{xk}(\theta_{k-1}) H[y_{xk}(\theta_{k-1}) \| h_\theta(k|x)] \quad \forall k \leq i \\
&\implies \theta_{i+1}^{(k)} = \theta_k^{(k)} \quad \forall k \leq i.
\end{aligned}
$$

Furthermore, assume that $i < K$. Because $\theta_i^{(1)}, \ldots, \theta_i^{(i)}$ are finite, we have $0 < w_{x(i+1)}, y_{x(i+1)} < 1$ and thus

$$
\theta_{i+1}^{(i+1)} \in \arg\min_\theta \sum_x \pi(x) w_{xk}(\theta_i) H[y_{xk}(\theta_i) \| h_\theta(i+1|x)]
$$

exists and is unique by A1. The statement follows by taking $\bar{Q}$ to be the matrix induced by the parameters $(\theta_1^{(1)}, \ldots, \theta_K^{(K)})$. $\qquad\square$

Finally, we show how TCSR relates to the fixed-point iteration with parametric approximation through an empirical Markov chain and state distribution.

**Proposition 3.** *Algorithm 1 is equivalent the fixed-point iteration* (5) *with the empirical Markov chain and state distribution*

$$p(x'|x) = \sum_m \mathbf{1}\{x_m = x, x'_m = x'\}/\sum_m \mathbf{1}\{x_m = x\}, \qquad \pi(x) = \sum_m \mathbf{1}\{x_m = x\}/M.$$

*Proof.* We begin as in the proof of Proposition 2 and rewrite the divergence minimized at each iteration of (5) in terms of binary cross-entropy as

$$d_\pi(P\overleftarrow{Q_{\theta'}}, Q_\theta) = \sum_x \pi(x) \sum_{k=1}^K w_{xk} H[y_{xk}\|h_\theta(k|x)],$$

$$y_{xk} = \sum_{x'} p(x'|x)h_{\theta'}(k-1|x'),$$

$$w_{xk} = \begin{cases} 1 & \text{if } k = 1, \\ \sum_{x'} p(x'|x)S_{\theta'}(k-2|x') & \text{if } k \geq 2. \end{cases}$$

Next, we plug in in the definition of $\pi(x)$ and $p(x'|x)$ and use the linearity of the binary cross-entropy to find that

$$
\begin{aligned}
d_\pi(P\overleftarrow{Q_{\theta'}}, Q_\theta) &= \sum_x \pi(x) \sum_{k=1}^K w_{xk} H[y_{xk}\|h_\theta(k|x)] \\
&= \frac{1}{M} \sum_{m=1}^M \sum_{k=1}^K w_{x_m k} H[y_{x_m k}\|h_\theta(k|x_m)] \\
&= \frac{1}{M} \sum_{m=1}^M \sum_{k=1}^K w_{x_m k} \sum_{x'} p(x' \mid x_m) H[h_{\theta'}(k-1|x')\|h_\theta(k|x_m)] \\
&= \frac{1}{M} \sum_{m=1}^M \sum_{k=1}^K w_{x_m k} H[h_{\theta'}(k-1|x'_m)\|h_\theta(k|x_m)] \\
&= \frac{1}{M} \sum_{m=1}^M \Bigg( H[\mathbf{1}\{x'_m = \varnothing\}\|h_\theta(k|x_m)] \\
&\qquad\qquad + \sum_{k=2}^K S_{\theta'}(k-2|x'_m) H[h_{\theta'}(k-1|x'_m)\|h_\theta(k|x_m)] \Bigg).
\end{aligned}
$$

The last equality establishes the equivalence between (5) and Line 8 in Algorithm 1. $\qquad\square$

## A.2 TCSR($\lambda$)

In this section, we discuss how the one-step temporal consistency equations (2) can be extended to multistep transitions, and we present an algorithm that takes advantage of this extension. Without loss of generality, we can assume that $p(\varnothing|\varnothing) = 1$, i.e., that the terminal state is absorbing. The $\ell$-hop transition probability $p_\ell(x'|x)$ is given by

$$p_1(x'|x) = p(x'|x), \qquad\qquad p_\ell(x'|x) = \sum_z p(x'|z)p_{\ell-1}(z|x), \quad \ell \geq 2.$$

The probability of reaching the terminal state, starting from $x$, on or before the $\ell$th step is given by $p_\ell(\varnothing|x)$. Using the Markov property and the law of total expectation, we can write a generalization of (2) as follows. For all $x$ and all $k \geq 1$, we have

$$f(k|x) = \mathbf{E}_{x' \sim p_\ell(\cdot|x)}[f(k - \ell|x')], \qquad\qquad S(k|x) = \mathbf{E}_{x' \sim p_\ell(\cdot|x)}[S(k - \ell|x')],$$

---

**Algorithm 2** TCSR($\lambda$)

---

**Require:** Sequences $\tilde{\mathcal{D}} = \{(x_{m0}, \ldots, x_{mt_m})\}$, initial parameters $\theta$, decay $\lambda \in [0, 1]$
 1: **repeat**
 2:    **for** $\ell = 1, \ldots, K$ **do**                $\triangleright$ Weights and targets for $\ell$-step lookahead.
 3:      **for** $m = 1, \ldots, M$ **do**
 4:        **for** $k = 1, \ldots, \ell$ **do**                    $\triangleright$ Based on observed outcomes.
 5:          $y_{mk}^{(\ell)} \leftarrow \mathbf{1}\{t_m = k \wedge x_{mt_m} = \varnothing\}$
 6:          $w_{mk}^{(\ell)} \leftarrow \mathbf{1}\{t_m \geq k\}$
 7:        **for** $k = \ell+1, \ldots, K$ **do**              $\triangleright$ Based on predictions at $x_\ell$.
 8:          $y_{mk}^{(\ell)} \leftarrow h_\theta(k-\ell|x_{m\ell})$
 9:          $w_{mk}^{(\ell)} \leftarrow S_\theta(k-\ell-1|x_{m\ell})$
10:      $c_\ell \leftarrow (1-\lambda)^{\mathbf{1}\{\ell < K\}} \lambda^\ell$
11:    $\theta \leftarrow \arg\min_\theta \sum_{\ell=1}^{K} c_\ell \sum_{m=1}^{M} \sum_{k=1}^{K} w_{mk}^{(\ell)} H[y_{mk}^{(\ell)} \| h_\theta(k|x_m)]$
12: **until** $\theta$ has converged

---

where we slightly abuse notation and denote, for all $k \geq 1$,

$$f(-k|x) = \begin{cases} 1 & \text{if } x = \varnothing, \\ 0 & \text{otherwise}, \end{cases} \qquad S(-k|x) = \begin{cases} 0 & \text{if } x = \varnothing, \\ 1 & \text{otherwise}. \end{cases}$$

We omit the multistep generalization of the fixed-point iterations (3) and (5), but we note that they can be obtained by repeatedly composing the transition and backshift operators (assuming that the definition of the latter is extended appropriately). Inspired by the LSTD($\lambda$) algorithm in the reinforcement learning literature [1], we present a generalization of Algorithm 1. We call it TCSR($\lambda$) and describe it in Algorithm 2.

The dataset $\tilde{\mathcal{D}}$ given as input to the algorithm consists of all subsequences of $\mathcal{D} = \{s_n\}$. This set extends the transitions $\mathcal{T} = \{(x_m, x'_m)\}$ we considered as input to TCSR in the main text and encompasses the full subsequence starting from every non-final state. The algorithm then repeatedly solves a survival regression problem (Line 11). The weights and targets are a combination of near-term observations and of predictions at future states that are $1, 2, \ldots, K$ steps away. The algorithm also uses an additional weight, $c_\ell$, that is associated with $\ell$-step lookahead targets, and that decreases exponentially fast with the lookahead. We identify two cases of special interest.

1. When $\lambda = 0$, only the next state is taken into account. In this case, one can verify that TCSR(0) is equivalent to Algorithm 1 presented in the main text.

2. When $\lambda = 1$, only the targets associated with the $K$-step lookahead are taken into account. In that case, TCSR(1) is identical to the landmarking estimator we discuss in the main text. That is, the algorithm is equivalent to the MLE (1) on the augmented dataset obtained by including intermediate states and the remaining time-to-event or time-to-censoring.

For values of $\lambda$ strictly between 0 and 1, TCSR($\lambda$) outputs models that can be understood to interpolate between these two extremes. A complete study of the benefits of choosing $0 < \lambda < 1$ is outside of the scope of this paper, but in the context of learning a scalar-valued function we note that the RL literature contains examples where such a choice leads to better sample-efficiency [7].

### A.3 Extensions to problems with actions

Our presentation so far has synthesized temporal difference learning—one of the central intellectual ideas in RL—and survival analysis. In RL, the algorithms SARSA and Q-learning naturally extend the insight of temporal difference learning to problems with actions. We provide a similar generalization here.

Consider a setting where data has been collected according to a baseline policy $\mu$, which assigns a probability $\mu(x, a)$ to action $a \in \{1, \cdots, A\}$ in state $x$. A transition kernel $p$ encodes the probability $p(x'|x, a)$ of transitioning from state $x$ to state $x'$ when action $a$ is chosen. Write

$$S^\mu(k|x) = [T > k \mid x_0 = x, a_0 = a, a_t \sim \mu(x_t) \ \forall t \geq 1]$$

---

**Algorithm 3** Survival SARSA for policy evaluation.

---

**Require:** dataset of actions and transitions under $\mu$ $\mathcal{T} = \{(x_m, a_m, x'_m, a'_m)\}$, initial parameters $\theta$

1: **repeat**
2:     Sample $(x, a, x', a') \sim \mathcal{T}$.
3:     $y_1 \leftarrow \mathbf{1}\{x' \neq \varnothing\}$                        $\triangleright$ Based on one-step observed outcome.
4:     $w_1 \leftarrow 1$
5:     **for** $k = 2, \ldots, K$ **do**                       $\triangleright$ Based on predictions at $x'$.
6:         $y_k \leftarrow h_\theta(k-1|x', a')$
7:         $w_k \leftarrow S_\theta(k-2|x', a')$
8:     $\theta \leftarrow \theta - \alpha \sum_{k=1}^{K} w_k \nabla_\theta H[y_k \| h_\theta(k|x, a)]$
9:     Update stepsize $\alpha$
10: **until** $\theta$ has converged

---

---

**Algorithm 4** Incremental temporally-consistent survival regression.

---

**Require:** dataset of transitions $\mathcal{T} = \{(x_m, x'_m)\}$, initial parameters $\theta$.

1: **repeat**
2:     $y_1 \leftarrow \mathbf{1}\{x' \neq \varnothing\}$                        $\triangleright$ Based on one-step observed outcome.
3:     $w_1 \leftarrow 1$
4:     **for** $k = 2, \ldots, K$ **do**                       $\triangleright$ Based on predictions at $x'$.
5:         $y_k \leftarrow h_\theta(k-1|x')$
6:         $w_k \leftarrow S_\theta(k-2|x')$
7:     $\theta \leftarrow \theta - \alpha \sum_{k=1}^{K} w_{mk} \nabla_\theta H[y_k \| h_\theta(k|x)]$
8:     Update stepsize $\alpha$
9: **until** $\theta$ has converged

---

for the probability that the time-to-event exceeds $k$ given the $a_0$ is chosen in state $x_0$ and the baseline policy is applied thereafter. As before, the survival function $S^\mu$ is related to the $k$-step hazard $h^\mu(k|x, a) = \mathbf{P}[T = k \mid T \geq k, x_0 = x, a_0 = a, a_t \sim \mu(x_t) \, \forall t \geq 1]$ through the formula $S^\mu(k|x) = \prod_{k'=1}^{k}[1 - h^\mu(k'|x, a)]$.

Algorithm 3 estimates a parametric approximation to $h^\mu$ using a batch of data collected by applying $\mu$. It is analogous to the online or incremental variant of temporally-consistent survival regression presented in Algorithm 4. In all places where Algorithm 4 accepted a state as input, Algorithm 3 takes a state-action pair. If Algorithms 1 and 4 are viewed as forms of temporal difference learning, then Algorithm 3 can be viewed as SARSA [7], which itself is a close relative of $Q$-learning.

The survival function is closely related to fundamental objects in dynamic programming and reinforcement learning. If a reward of 1 were earned in each period before an event occurs, it is typical to write the future expected reward as

$$\mathbb{Q}^\mu(x, a) = \mathbf{E}\left[\sum_{k=1}^{K} \mathbf{1}(x_k \neq \varnothing) \mid x_0 = x, a_0, x_t \sim \mu(x_t, \cdot) \, \forall t \geq 0\right] = \sum_{k=1}^{K} S^\mu(k|x, a). \quad (6)$$

Through such a formula, estimates of the survival function or hazard induce estimates of the value function $\mathbb{Q}$, but the former objects are an ideal fit for time-to-event data and have been extensively studied for decades.

If $\mathbb{Q}^\mu$ were estimated accurately, then improved performance can be attained through the policy $\mu^+$ which, when faced with any state $x$, chooses $\arg\max_a \mathbb{Q}^\mu(x, a)$. Many RL algorithms mimic the classic policy iteration algorithm by interleaving steps of policy evaluation and policy improvement of this form. The objective in (6) aligns naturally with problems where the goal is to retain a customer, to keep a from developing a condition, or to prevent a critical machine from breaking, for as long as possible.

# B  Experimental evaluation

In this appendix, we provide additional details on the experiments presented in Section 4 of the main text. Note that the supplementary material also contains the code necessary to reproduce all the experiments presented in the paper. The code is structured as follows.

- A Python library called `tdsurv` provides an implementation of TCSR($\lambda$), the algorithm presented in Appendix A.2. This algorithm generalizes all the approaches investigated in the paper (including TCSR, initial state and landmarking) and underpins all experimental results. The library implements the Cox PH model and several other parametric models. New models can be added easily, by specifying a function that gives the hazard log-odds.

- A set of computational notebooks (Jupyter notebook files) documents the experiments in detail. They rely on the `tdsurv` library, and they can be used to reproduce experimental results and figures presented in the paper.

Installation instructions and technical details can be found in the `README` file included with the code archive. We plan to open-source the library and the notebooks on Github upon publication of the paper.

## B.1  Metrics

Devising evaluation metrics for survival models is challenging due to the presence of right-censored data; Care must be taken to properly account for censoring and to avoid bias. We describe in more detail the four metrics that we use in the main text and in section B.3 of this appendix.

**Concordance index.** This metric measures the agreement between the ranking induced by the observed time-to-event and that induced by a set of scores. Random scores achieve a CI of $0.5$, while a CI of $1$ corresponds to perfect agreement between scores and observed time-to-event. In the Cox-PH model, a natural choice is to score instances by the partial hazard log-odds $\beta^\top \phi_{x_0}$. The concordance index (CI) is similar to the Kendall-tau rank correlation: it counts the fraction of pairs whose relative order under the true and predicted ranking is identical. The CI accounts for right-censored observations by removing pairs that are incomparable. A precise definition is given in [4], and we also refer the interested reader to Raykar et al. [6]. In our experiments, we use the implementation provided by the `lifelines` Python library.[1]

**RMSE.** This metric measures the $\ell_2$-distance between parameter estimate $\beta$ and a gold-standard value $\beta^\star$, defined as $\|\beta - \beta^\star\|_2$. This metric can be thought of as a proxy to measuring, e.g., how well a model estimates the effect of a treatment on survival.

**Integrated Brier score.** This metric is an extension of the well-known Brier score to the survival setting. Given a validation set $\mathcal{D} = \{(x_{n0}, t_n, c_n) : n = 1, \dots, N\}$, it is defined as

$$\frac{1}{K} \sum_{k=1}^{K} \sum_{n=1}^{N} \left\{ \frac{[0 - S(k|x_{n0})]^2}{w(t_n)} \mathbf{1}\{t_n \leq k \wedge c_n = 0\} + \frac{[1 - S(k|x_{n0})]^2}{w(k)} \mathbf{1}\{t_n > k\} \right\},$$

where $w(k)$ is the Kaplan-Meier estimate of time-to-censoring. It favors survival probabilities that are accurate and calibrated uniformly at all horizons. See Graf et al. [3] and Gerds and Schumacher [2] for more details.

**Predictive log-likelihod.** This metric captures the likelihood of the model on a validation dataset. It is defined as

$$\sum_{n=1}^{N} \left[ \mathbf{1}\{c_n = 0\} \log f(t_n|x_{n0}) + \mathbf{1}\{c_n = 1\} \log S(t_n|x_{n0}) \right].$$

Similarly to the integrated Brier score, it favors accurate and well-calibrated survival distributions.

---

[1]See: `https://lifelines.readthedocs.io/en/latest/lifelines.utils.html`.

Throughout the experiments, we add a small $\ell_2$ penalty on the model parameters $\theta$ to the optimization objective of the various learning algorithms. All but one metric are sensitive to the choice of regularization strength (the concordance index is the only metric that is usually highest with models trained without regularization). For this reason, we run a grid search across a range of regularization strengths, and report the best result for each combination of dataset, metric and learning algorithm. Details on the grid search are presented in the `*-experiments.ipynb` notebooks provided with the code.

## B.2 Datasets

The two clinical datasets we discuss in the main text are lightly preprocessed. In the PBC2 dataset, some biomarkers are absent for some of the measurements. We replace missing values with the median of the all the values of the corresponding biomarker; Lee et al. [5] use a similar approach but use the mean instead of the median. In the AIDS dataset, position in the sequence is predictive of survival, controlling for the features. We address this by augmenting the state representation $\phi_x$ with an additional dimension containing the index of the observation within the sequence. More details on the datasets and preprocessing steps are given in the `*-exploratory-analysis.ipynb` notebooks provided with the code.

**Random walk dataset.** We experiment with the following generative model of sequences. Denote by $\mathcal{N}(\mu, r^2 I)$ a multivariate Gaussian distribution with mean $\mu$ and spherical covariance of radius $r$, and let $\sigma(u) = 1/(1+e^{-u})$ be the logistic function. We initialize each sequence with a 20-dimensional state $x_0 \sim \mathcal{N}(0, r_0^2 I)$. For $\ell = 1, 2, \ldots$, we sample $z_\ell \sim \text{Bernoulli}(p_i)$ with $p_i = \sigma(\gamma^\top x_{\ell-1} + b)$. If $z_\ell = 1$, the sequence terminates and we set $x_\ell = \varnothing$. Otherwise, we sample $x_\ell \sim \mathcal{N}(x_{\ell-1}, w^2 I)$. We repeat this process until the sequence terminates or until $\ell = 10$. By construction, this process is Markovian. Despite its simplicity, the sequence dynamics (conditioned on survival) and the survival distribution are non-trivial; Informally, the process "moves away" from the point $\gamma$ conditioned on survival. We instantiate this model with

$$r_0^2 = 1.0, \qquad w^2 = 0.5, \qquad \gamma \sim \mathcal{N}(0, I), \qquad b = -3.$$

We sample 5 independent training sets containing 200 sequences each, and we also sample a validation set consisting of $10\,000$ sequences. Approximately $24\,\%$ of sequences are right-censored. We define the gold-standard $\beta^\star$ as the MLE (1) obtained on the validation set. We set $\phi_x = x$ and train Cox PH models using increasingly larger subsets of the training data. In the results presented in the main text and Section B.3 of this appendix, we report the performance on the validation set, averaged over the 5 training splits.

## B.3 Additional experimental results

In Figure 1, we report the performance of the three approaches discussed in the main text on two additional metrics that capture models' ability to estimate accurate, calibrated survival probabilities. These results are consistent with the observations we made based on the performance in terms of RMSE and concordance index. For PBC2 and RW, our algorithm (TCSR) is *a*) usually more sample-efficient, and *b*) it achieves better overall performance given the full data. On the AIDS dataset, there is no significant advantage obtained by landmarking and TCSR, an observation we explain by the absence of rich sequence dynamics compared to the other two datasets.

**Computational setup.** The experiments reported in the paper were run on a Google cloud `c2-standard-16` instance with 16 vCPUs and 64GB RAM. Note that they have also been replicated on a 2019 laptop with 4 cores and 16GB RAM, with a minimal difference in running time. Overall, on the datasets that we consider, all learning algorithms converge in a fraction of a second in most cases, and computational infrastructure & running time are likely not a concern for reproducibility.