# OpenReview forum: "Temporally-Consistent Survival Analysis"
_NeurIPS.cc/2022/Conference — NeurIPS 2022 Accept_

### Official Review · Reviewer_hyyi · 2022-06-30

**Rating:** 7
**Confidence:** 4
**Soundness:** 3 good
**Presentation:** 3 good
**Contribution:** 4 excellent

**Summary:**

This paper presents a new algorithm for survival analysis with time-varying features.  This paper exploits a similarity between survival analysis and reinforcement learning, and the proposed algorithm can be seen as a natural adaptation of the temporal-difference learning in reinforcement learning to the survival analysis.


**Questions:**

I have no questions.


**Limitations:**

Limitations are not clearly stated.  Lines 227-230 in Section 3.3 might describe the limitation, but I could not find any description on limitation in Section 4.


**Strengths And Weaknesses:**

== Strengths ==
+ The major contribution of this paper is to establish the connection between the survival analsysis with time-variying features and the reinforcement learning.  By using the connection, this paper shows a new algorithm TCSR, which is based on the temporal-difference learning in reinforcement learning.  I think that this contribution is beneficial for the community of survival analysis.
+ Related to the previous point, this paper shows a new approach for survival analysis with time-varying features.  Usually, we estimate the outcome for a single state $x$ in survival analysis.  However, this paper shows that it is better to estimate the difference of outcomes between two consecutive states $x$ and $x'$.  This idea is known as temporal-difference learning in reinforcement learning, but it is a new idea in survival analysis.


== Weakness ==
+ Propositions 1-3 supports the superiority of the new algorithm TCSR over the known algorithm (i.e., Landmarking) in the experiments described in Section 4.1.  However, these propositions have less relationship with the experimental results described in Section 4.2, because each state $x$ appears only once in the datasets used in Section 4.2.  The difference comes from the fact that discrete states were used in Section 4.1 while the states are created based on feature vectors with continuous values.  I think, this paper implicitly assumes that, if two states $x$ and $x'$ are similar (i.e., $\phi_{x} \approx \phi_{x'}$), then the true survival distribution $q_{x}$ and $q_{x'}$ are similar.  However, this paper does not state this assumption explicitly.
+ The experiments described in Section 4.1 can be improved.  Since this paper uses a synthetic dataset in Section 4.1, we can calculate the true probability distribution $q_{x}$ for each state $x$.  Therefore, we can evaluate the outputs of the algorithms by directly comparing with the true probability distribution.  I think that the Kendall-$\tau$ (and C-index) is an appropriate metric only when we do not know the true probability distribution.
+ The presentation of this paper can be improved.  For example, $x_{t}$ in Lines 91-92 is not defined (and its definition seems different from $x_{t}$ defined in Lines 97-100).  For another example, $c_{t}$ in Line 122 is undefined, although $c$ in Line 123 is defined in Line 100.  In equations between Lines 51-52 in Section A.1, the $x$ in the third line and $x'_{m}$ in the fourth line are undefined.
+ I think there are some errors in mathematics.  In Lines 27-28 in Section A.1, the sum over $K=1,2,\ldots,K+1$ in the second term of the second equality should be $K=2,3,\ldots,K+1$.  In Lines 28-29 in Section A.1, $w_{xk}(\theta')$ must be zero when $k \geq 2$, because $h_{\theta'}(\ell | x') = 1$ when $\ell=0$ by the definition of $h$ in Lines 93-94 in Section 2.


== Minor issues ==
+ It is better to clarify that $T$ in Line 87 is discrete.
+ I am not sure if it is appropriate to use the terminology "converges" in Proposition1 (and Proposition 2).  Proposition 1 only shows how to compute $Q$ from the transition probabilities $p(x'|x)$ by using dynamic programming assuming that we know the true transition probabilities.
+ There are many grammatical errors (e.g., "is reaches" -> "it reaches" in Line 99, "describe" -> "describes" in Line 113, and "equivalent" -> "equivalent to" in Line 213).
+ I think that it is better to rephrase "estimate survival" with "estimate survival distribution" in Line 94.  (Note also that similar phrases appear several times after Line 94.)

---

> ### Author Response · Authors · 2022-08-02
> **Response to Reviewer hyyi**
>
> Thank you very much for your thorough review! We are grateful for such detailed feedback, and we will make a number of improvements based on your comments, including:
>
> - Clarifying the relation between discrete states (Section 3) and continuous, vector-valued state representations combined with parametric models that make an implicit smoothness assumption (experiments in Section 4.2).
> - Introducing the notation more systematically and fix a few inconsistencies (e.g., $x_t$ on line 91 vs. on line 100, $c$ and $c_t$, and some of the notation in the appendix).
> - Fixing the grammatical errors and other minor issues that you point out.
>
> We briefly comment on three points that you raise.
>
> > Since this paper uses a synthetic dataset in Section 4.1, we can calculate the true probability distribution $q_x$ for each state $x$. Therefore, we can evaluate the outputs of the algorithms by directly comparing with the true probability distribution.
>
> In Section 4.1, we chose a ranking metric because we wanted to illustrate the models’ ability to discriminate “good” treatments from “bad” ones. In practice, this is often an important goal (think of, e.g., randomized control trials). Nevertheless, we will take your suggestion into account, and we will consider additional metrics that quantify the quality of the probability distribution.
>
> > In Lines 27-28 in Section A.1, the sum over $k = 1,2,…,K+1$ in the second term of the second equality should be $k= 2,3,…,K+1$
>
> We agree that starting that sum from $k = 2$ is easier on the reader, and we will make that change in the final version. We believe that the current version is technically correct, since the inner sum $\sum_{k’=1}^{k-1} \cdots$ evaluates to zero when $k = 1$, but your suggestion is clearer.
>
> > In Lines 28-29 in Section A.1, $w_{xk}(\theta')$ must be zero when $k≥2$, because $h_{\theta'}(\ell \vert x')=1$ when $\ell = 0$ by the definition of h in Lines 93-94 in Section 2
>
> Following our definition in Section 2 (l. 93), $h(0 \vert x’) = 1$ only if $x’ = \varnothing$. For any $x’ \ne \varnothing$, we have $h(0 \vert x’) = 0$, and as a consequence $w_{xk}(\theta)$ is not necessarily equal to zero for $k \ge 2$.
>
> Alternatively, we could have written:
> $$w_{xk}(\theta) = \sum_{x’ \ne \varnothing} p(x’ \vert x) \prod_{\ell = 1}^{k - 2}[1 - h_{\theta’}(\ell \vert x’)], $$
> for $k \ge 2$.
>
> Once again, we thank you for your careful proofreading and constructive feedback.

---

> > ### Comment · Reviewer_hyyi · 2022-08-09
> > **Thank you**
> >
> > Regarding the last point, you are right and I was wrong, though I prefer the alternative one.

---

### Official Review · Reviewer_mhhX · 2022-07-05

**Rating:** 7
**Confidence:** 3
**Soundness:** 3 good
**Presentation:** 3 good
**Contribution:** 3 good

**Summary:**

The authors borrow the idea from temporal difference learning to model sequential survival analysis with the temporal consistency. The authors discretize the time and covariates and prove exact solutions with dynamic programming and also provide approximate solutions with parametric models. The authors show that their method has sample efficiency and better predictive performance on empirical studies.

**Questions:**

1. How do the authors handle continuous covariates in the real-world studies?
2. Survival times are usually continuous. Can the authors' model handle continuous times?
3. At each time step, we may have a history of covariates (covariates in previous time steps). How do the authors' framework handle the history of covariates? Maybe using the hidden state of a RNN as a continuous state?

**Limitations:**

The proposed models assume the number of states is finite. This may be my misunderstanding. Feel free to correct me.

**Strengths And Weaknesses:**

Strengths:
1. The authors provide a framework to model temporal consistency in sequential survival predictions.
2. The authors describe the problem in a sequential decision way and provide dynamic programming and parametric solutions. I think the solution is elegant. I personally thought about the problem before and got confused on RNN models without temporal consistency. The authors here solve my concerns with a simple setting.
3. The authors show the benefit on both sample efficiency and better predictive performance on three datasets.
4. The writing is clear to me.

Weakness:
1. There is one thing that confuses me. It looks that the authors use finite number of states in model description. But in real world, the features are usually continuous so we have infinite amount of states. How do the authors handle continuous covariates in the real-world studies?

---

> ### Author Response · Authors · 2022-08-02
> **Response to Reviewer mhhX**
>
> Thank you for your review and your support of our paper! We briefly address the three questions that you raise.
>
> > How do the authors handle continuous covariates in the real-world studies
>
> We assume that the state space is finite only to describe that, in principle, one could compute the exact survival distribution by dynamic programming. A presentation with continuous states would introduce many technicalities that are orthogonal to the main ideas there.
>
> After presenting an exact result for finite state spaces, we introduce an algorithm that uses parametric approximations. That algorithm works with continuous state representations, without any modifications. We are sorry for the confusion here, and will emphasize more clearly that, in practice, the assumption of a finite state space is not necessary to use our algorithm.
>
> > Survival times are usually continuous. Can the authors' model handle continuous times?
>
> It’s worth noting that some important applications involve discrete survival times. For example, e.g. user/customer lifetimes are usually defined in terms of daily activity metrics, or billing cycles.
>
> Nevertheless, we believe that it could be possible to extend our method to handle continuous-time data (we mention this as a possible direction for future work in Section 5). We believe that such an extension would involve two steps:
>
> 1. Extend our discrete-time model to handle missing observations (a.k.a, irregularly-sampled covariates). This would likely entail an argument similar to that in Section A.2 in the appendix, where we develop multistep lookaheads.
> 2. Given the ability to cope with missing data, we can consider increasingly fine-grained discretizations. In the limit, we obtain a continuous-time model.
>
> This is only a sketch, and a rigorous development would likely require working out non-trivial technical details, which is outside the scope of our submission. We feel that the main insights of our paper—about the important statistical benefits of enforcing temporal consistency in survival modeling—are just as relevant in continuous time models, and hope our submission encourages this follow-on work.
>
> > How do the authors' framework handle the history of covariates? Maybe using the hidden state of a RNN as a continuous state?
>
> Our framework is based on a Markov model, in which case the covariates at time t contain all pertinent information about the past covariates. This requires a good feature representation. In the applications in Section 4.2, it seems the basic handcrafted covariates used in past work are effective, without any modification.
>
> In other applications, what you mention seems like the right approach: one could train an RNN on some subset of the data and use the hidden state as a feature representation.

---

> > ### Comment · Reviewer_mhhX · 2022-08-06
> > **The authors have addressed my questions**
> >
> > The authors have addressed my questions and there are no further concerns.

---

### Official Review · Reviewer_fEGu · 2022-07-10

**Rating:** 7
**Confidence:** 4
**Soundness:** 4 excellent
**Presentation:** 3 good
**Contribution:** 3 good

**Summary:**

The authors propose a novel algorithm to estimate a discrete-time survival model by exploiting a temporal-consistency condition inspired by connections with literature on reinforcement learning. Focusing on survival analysis in the dynamic setting, this work proposes a theoretical framework together with a practical algorithm that fits a parametric survival model by using sample trajectories. At its core, this work argues in favor of the idea that temporal consistency can dramatically increase statistical precision within survival models. Empirically, the authors provide an evaluation of two real-world and two synthetic datasets.


**Questions:**

Mentioned in the main body of the review.

**Limitations:**

The authors address the limitation of their work at a relatively high level.

**Strengths And Weaknesses:**

Originality:
To the best of my knowledge, the authors propose a novel algorithm introducing an interesting methodological advancement in the field of survival analysis. The authors clearly acknowledge related literature by highlighting common challenges in the field and how their contributions are able to fill these gaps. Clear introduction, interesting idea, and nice organization.
I also appreciated the parallelism with RL methods, both in Section 3.4 and the appendix.

Quality:
The authors clearly made an effort to accurately describe the general setting and methodology. Overall, I find the idea quite elegant in its simplicity. Below are some comments/questions which I'd like the authors to address (in no particular order):

- I feel like the empirical analysis is characterized by a rather short discussion as opposed to the methodological contribution which is very clearly explained. Therefore, I believe the manuscript would gain from a qualitative discussion of the results (whenever possible): what should the reader take away from the experiments? how do these results confirm the theoretical contributions? For example, when speaking of quantitative results, the authors don't explain why the performance of TCSR degrades with a larger number of sequences (Figure 2, bottom left). Is this only due to randomness? Why does it seem to be so consistent in the PBC2 dataset? (i.e., across all k-folds). I believe that elaborating on these extreme cases (both positive and negative) can elevate the quality of the entire paper.
- In Figure 2, the authors show the performance of the three implemented methods across three datasets. Is there a reason why the "#sequences" stops at 200? I might be misunderstanding something, but from Table 1, it seems like both real-world datasets could potentially allow for more than 200 sequences (even considering the implemented k-fold cross-validation)
- In Section 4.2, the authors use the MLE estimate for the regression coefficients as gold-standard, which is clearly a valid approach. To bring this reasoning to an extreme, would it be possible to generate a survival analysis dataset from a set of *known* parameters and check whether these methods are able to converge to the "absolute-true" values? If possible, I believe this scenario would greatly benefit the manuscript.

Clarity:
The paper is well organized and easy to read. All visualizations are clear and effective.

Significance:
The authors address an interesting problem, with novel ideas. The approach shows clear theoretical and practical advantages compared to traditional methods. However, I believe the current version of the paper could be more effective in justifying the contributions through a more detailed experiment analysis.

---

> ### Author Response · Authors · 2022-08-02
> **Response to Reviewer fEGu**
>
> Thank you for your review and your support of our paper! We will take your suggestions into account to improve future versions of the manuscript. In particular, given more space, we will expand on the take-aways from Sections 4.1 and 4.2 and link them back to Section 3. We briefly answer some of the specific questions that you raise.
>
> > why the performance of TCSR degrades with a larger number of sequences (Figure 2, bottom left). Is this only due to randomness? Why does it seem to be so consistent in the PBC2 dataset? (i.e., across all k-folds)
>
> This appears to be noise. We have re-run the PBC2 experiment with additional random seeds (setting `seed` to 1, 2, 3, .. in cell 6 of pbc-final.ipynb), and the RMSE no longer degrades at around 100 samples.
>
> > In Figure 2, the authors show the performance of the three implemented methods across three datasets. Is there a reason why the "#sequences" stops at 200?
>
> In Figure 2, we wanted to highlight the performance in the low-data regime and decided to keep the same x-axis for ease of comparison across datasets. In addition, recall that the gold-standard $\beta^\star$ is estimated using the _initial state_ method on the full dataset (and not only on the validation set). As such, it becomes delicate to interpret the RMSE as the training set approaches the full dataset (for example, the RMSE of the _initial state_ approach would go to zero by construction).
>
> > To bring this reasoning to an extreme, would it be possible to generate a survival analysis dataset from a set of known parameters and check whether these methods are able to converge to the "absolute-true" values?
>
> This is an interesting suggestion. To some extent, we do this with the Random Walk dataset. Given that the data comes from a simulation, we can generate as many traces as we want, and in this case we define $\beta^\star$ to be the MLE on a much larger & independent dataset of 10k samples (details are provided in lines 179-184 in the Appendix). As such, $\beta^\star$ is close to some notion of “absolute-truth” in this case. However, the true survival distribution does not follow a Cox-PH model exactly, and so $\beta^\star$ is “only” the best Cox-PH approximation. We have yet to find a generative model that has both (a) Markovian dynamics and (b) a survival distribution that matches a Cox-PH model exactly from every state.

---

### Official Review · Reviewer_VqoZ · 2022-07-20

**Rating:** 6
**Confidence:** 5
**Soundness:** 3 good
**Presentation:** 3 good
**Contribution:** 3 good

**Summary:**

The authors formulated the dynamic survival analysis problem as modeling the distribution of hitting time for a terminal state given sequences of states. Under the assumption that the sequences of states follow a Markov chain, the survival distribution is determined by the transition matrix and can be computed by a simple recursive algorithm once the transition matrix is known. Without the access to transition matrix, the authors developed a temporally-consistent survival regression (TCSR) algorithm to estimate parametric models given sequences of states. The authors also compared the proposed method with the model only using initial-state information and landmark approach on both synthetic and real-world datasets.

**Questions:**

Please see the weaknesses above.

**Limitations:**

Yes, the authors have discussed limitations.

**Strengths And Weaknesses:**

Strengths:
* The authors made a very interesting connection between dynamic survival analysis and reinforcement learning. These insights can help take advantage of algorithms studied in reinforcement learning to estimate survival models.
* The manuscript is well written and provides the necessary background for general audiences, especially presenting the method step by step from a known transition matrix with nonparametric models, to a known transition matrix with parametric approximation, and finally to an unknown transition matrix.
* Exploiting the temporal consistency helps improve data efficiency as shown in the experiments.

Weaknesses:
* The proposed method requires successive measurements of covariates at regular time points and finite state space, which is not often the case in practice. The authors have discussed the potential extension to irregularly-sampled, continuous-time sequences. Could the authors also comment on how generalizable the method is to continuous state space as the covariates can be continuous random variables?
* Another important approach in dynamic survival analysis, time-dependent Cox regression, is missing in the discussion and comparison. Time-dependent Cox regression is different from both landmarking and joint modeling. Could the authors discuss and compare it with the proposed method?
* How accurate the unique fixed point $\bar{Q}$ in Proposition 2 is to the true survival matrix?
* I don't think the proposed method is completely complementary to existing dynamic survival models. It does make an additional assumption, Markov chain, so as the temporal consistency. If I missed anything here, could the authors explicitly explain what's the impact of the Makov chain assumption on characterizing the survival distribution?
* Definition of the event time is not clear in line 91: I presume T=k means that the sequence reaches the terminal state at (k+t)-th step given that x_t=x at the t-th step? Because x_t is not defined until line 98.

---

> ### Author Response · Authors · 2022-08-02
> **Response to Reviewer VqoZ**
>
> Thank you for your constructive review! Your feedback raises relevant questions, which we address below.
>
> > Could the authors also comment on how generalizable the method is to continuous state space as the covariates can be continuous random variables?
>
> In Section 3, we begin by assuming that the state space is finite. We do so only to describe that, in principle, one could compute the exact survival distribution by dynamic programming. A presentation with continuous states would introduce many technicalities that are orthogonal to the main ideas there.
>
> We then bring in the idea of using a parametric approximation and eventually introduce Algorithm 1 (TCSR). In practice, the assumption of a finite state space is not necessary to use TCSR: By using smooth parametric functions, the algorithm works with continuous state representations without any modifications.
>
> We apologize for the confusion (this point was mentioned by other reviewers as well) and will describe this in greater detail in a future version.
>
> > Could the authors discuss and compare [time-dependent Cox regression] with the proposed method?
>
> Time-dependent Cox regression is indeed a useful method to analyze dynamic survival data. In the discrete-time case, re-using the notation from our paper, we can write the corresponding model as
>
> $$\mathrm{logit}[h(k \vert x_0, x_1, \ldots)] = \beta^\top \phi_{x_k} + \alpha_k$$
>
> Note that the hazard at step $k$ depends on the state at step $k$. As such, time-dependent Cox regression is not a predictive model: given only the initial observation $x_0$, it is not possible to predict the distribution of time-to-event. (Doing so would entail learning an additional model that predicts possible futures $x_1, x_2, \ldots$, as is done in joint modeling.)
>
> In contrast, models obtained through landmarking or TCSR are predictive models. Given that our evaluation metrics (both in the main text and in the appendix) measure the models’ predictive performance, we cannot provide a meaningful comparison to time-dependent Cox regression.
>
> > How accurate the unique fixed point $\bar{Q}$ in Proposition 2 is to the true survival matrix?
>
> In proposition 2, $\bar{Q}$ minimizes $d_\pi(\bar{Q}, P \overleftarrow{\bar{Q}})$. Your comment highlights a more general open question around the study of temporal-difference methods. TD generally minimizes a notion of temporal inconsistency, and not a more common notion like prediction error.
>
> In the context of learning a value function in discounted MDPs, Theorem 1 part 4 here provides an error bound with linear models: https://www.mit.edu/~jnt/Papers/J063-97-bvr-td.pdf
>
> In general, it is an open question to characterize theoretically how well TD performs when nonlinear function approximation is used and there are inherent errors due to parametric approximation. We will add comments about these open questions in the literature.
>
> > If I missed anything here, could the authors explicitly explain what's the impact of the Makov chain assumption on characterizing the survival distribution?
>
> The inductive biases of our methods help if the sequences have Markov dynamics, but they could indeed hurt predictive performance if the sequences are strongly non-Markovian.
>
> We have experimented extensively with a variant of our algorithm that we present in Appendix A.2, TCSR($\lambda$). Informally, this variant lets us vary the level of temporal-consistency we enforce in the output. Interestingly, we have found that the maximum level of temporal-consistency ($\lambda = 0$, corresponding to Algorithm 1 in the main text) gives the best results on the datasets we consider. This suggests that the benefits of enforcing temporal-consistency may often outweigh the bias due to deviations from the Markov property in practice.
>
> > I presume T=k means that the sequence reaches the terminal state at (k+t)-th step given that x_t=x at the t-th step? Because x_t is not defined until line 98.
>
> You are correct: $\mathbf{P}[T = k \mid x_t = x]$ is the probability that the sequence reaches the terminal state at the $(k+t)$th step given that the sequence is at state $x$ at the $t$th step. We will introduce the notation more systematically and clarify this in the next version of the manuscript.

---

### Meta-Review · Area_Chair_pWsM · 2022-08-27

**Recommendation:** Accept
**Confidence:** Certain

**Metareview:**

This paper draws connections between survival analysis where covariates appear over time and td-learning in reinforcement learning. The connection is based on a recursion in the survival distribution when the process satisfies a Markov assumption. The paper then uses this connection to develop a new estimator for survival analysis using dynamic programming. The paper has the potential to connect two disparate communities in ML. From a technical standpoint, I am interested in what happens when the Markov assumption is weakened.

**Award:**

No

---

### Decision · Program_Chairs · 2022-09-14

Accept